# Pfaffian invariant identifies magnetic obstructed atomic insulators

Isidora Araya Day [1, 2, *], Anastasiia Varentcova [2], Dániel Varjas [1, 2, 3] and Anton R. Akhmerov [2]

**1** QuTech, Delft University of Technology, Delft 2600 GA, The Netherlands
**2** Kavli Institute of Nanoscience, Delft University of Technology, P.O. Box 4056, 2600 GA Delft, The Netherlands
**3** Department of Physics, Stockholm University, AlbaNova University Center, 106 91 Stockholm, Sweden

* i.araya.day@gmail.com

June 1, 2023

## Abstract

**We derive a $\mathbb{Z}_4$ topological invariant that extends beyond symmetry eigenvalues and Wilson loops and classifies two-dimensional insulators with a $C_4\mathcal{T}$ symmetry. To formulate this invariant, we consider an irreducible Brillouin zone and constrain the spectrum of the open Wilson lines that compose its boundary. We fix the gauge ambiguity of the Wilson lines by using the Pfaffian at high symmetry momenta. As a result, we distinguish the four $C_4\mathcal{T}$-protected atomic insulators, each of which is adiabatically connected to a different atomic limit. We establish the correspondence between the invariant and the obstructed phases by constructing both the atomic limit Hamiltonians and a $C_4\mathcal{T}$-symmetric model that interpolates between them. The phase diagram shows that $C_4\mathcal{T}$ insulators allow $\pm 1$ and 2 changes of the invariant, where the latter is overlooked by symmetry indicators.**

See also: online presentation recording

Topological crystalline insulators are phases of matter where it is impossible to define exponentially localized Wannier functions that respect the crystalline symmetries [1, 2]. Obstructed atomic insulators, on the contrary, allow symmetric and exponentially localized Wannier functions whose centers occupy maximal Wyckoff positions, such that a continuous and symmetric deformation cannot move them [3–5]. The symmetry representations of the occupied orbitals at high symmetry momenta—symmetry indicators [6–8]—distinguish part of the obstructed atomic insulators, but not all [9]. Reference [9] constructed Berry phase-based topological invariants that distinguish these phases in specific examples and put forward the conjecture of the universality of this approach.

Two-dimensional magnetic insulators belonging to the magnetic plane group $p4'$ that are symmetric under the product of four-fold rotation ($C_4$) and spinful time-reversal symmetry ($\mathcal{T}$) support distinct obstructed atomic insulating phases. A general Wyckoff position has an orbit of size four, hence a crystal with two occupied orbitals must have the Wannier centers located at maximal Wyckoff positions. This restriction allows the four distinct phases labeled by $\nu$ shown in Fig. 1: a spin singlet in the center or the corner of the Wigner-Seitz unit cell, and two phases with $z$-oriented spins located at the middle of the unit cell edge. The product $\delta$ of the eigenvalues of $C_2 = (C_4\mathcal{T})^2$ at the $C_2$-invariant momenta $X = (\pi, 0)$ or $Y = (0, \pi)$ differentiate the singlet phases from the spin-polarized ones [10]. However, even the full set of symmetry indicators only

provides an incomplete topological classification: phases $\nu = 0$ and $\nu = 2$ have identical representation content at every high-symmetry momentum. The crystalline symmetry guarantees that all the Wilson loops along reciprocal lattice vectors provide the same information as the symmetry indicators, and therefore distinguishing all four phases requires extending the approach of Ref. [9] to construct the topological invariant.

Our main result is a topological invariant $\nu$ that captures all the obstructed phases in a $C_4\mathcal{T}$-symmetric two-dimensional magnetic insulator. We identify the invariant by constructing a discrete quantity that utilizes the symmetry constraints on the wave functions, following a reasoning similar to the $\mathbb{Z}_2$ invariant in topological insulators [6, 11, 12]. Instead of the time-reversal symmetry operator $\mathcal{T}$, we use the operator

$$\Theta = \frac{C_4\mathcal{T} - (C_4\mathcal{T})^{-1}}{\sqrt{2}},\tag{1}$$

that protects the Kramers-like pairs at the high symmetry momenta $\Gamma = (0,0)$ and $M = (\pi, \pi)$ [13, 14]. This definition of $\Theta$ is different from $\Theta = (C_4\mathcal{T} + C_4^{-1}\mathcal{T})/\sqrt{2}$ used in Refs. [13, 14], which relies on using the operators $C_4$ and $\mathcal{T}$ absent within the symmetry group, but it is equivalent otherwise.

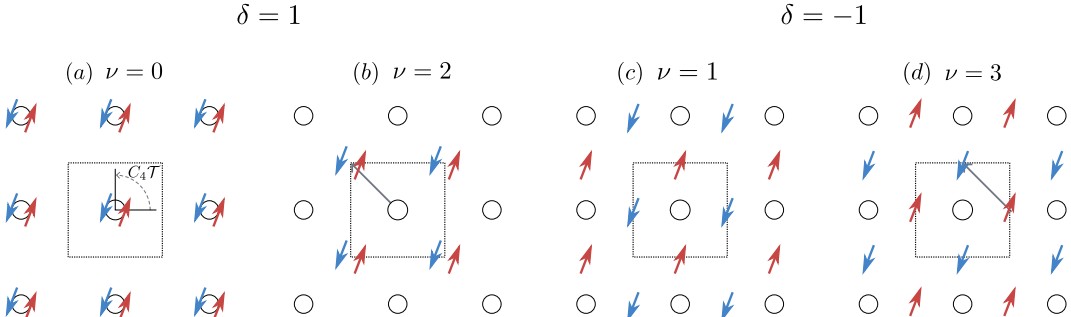

Figure 1: Different atomic limits of the $C_4\mathcal{T}$-symmetric insulator. The system is made out of atoms (empty circles) placed in the center of the Wigner-Seitz unit cell (square). The $C_4\mathcal{T}$ symmetry (grey dashed arrow) rotates the system by 90° around the atom and flips the spins. The four distinct atomic insulators are: (a) spin singlet located on the atom, (b) spin singlet at the corner of the unit cell, and (c-d) spins pointing in $\pm z$-direction. The two spins (red/blue) in a unit cell are of different orbital characters in all panels. The phases shown in panels (b) and (d) are related to the ones in panels (a) and (c) by a fractional lattice translation (grey solid arrows).

To exploit the symmetry, we formulate the invariant by using the occupied states only in the irreducible Brillouin zone (IBZ). Without loss of generality, we choose the irreducible Brillouin zone shown in Fig. 2, with the boundary path $\Gamma \to M \to X \to M \to \Gamma$. Stokes' theorem applied to the IBZ equates the Berry flux to the boundary Berry phase:

$$\int_{\text{IBZ}} \text{tr}\,\mathcal{F}\,d\mathbf{k}^2 - \oint_{\partial\text{IBZ}} \text{tr}\,\mathcal{A}\,d\mathbf{k} = 0 \mod 2\pi,\tag{2}$$

where $\mathcal{A}_{mn}(\mathbf{k}) = i\langle m\mathbf{k}|\partial_{\mathbf{k}}|n\mathbf{k}\rangle$ is the non-Abelian Berry connection, $\mathcal{F}_{mn}(\mathbf{k}) = \nabla \times \mathcal{A}_{mn}(\mathbf{k})$ is the non-Abelian Berry curvature, and $|n\mathbf{k}\rangle$ is an orthonormal basis of the occupied eigenstates of the Bloch Hamiltonian. Since the Berry connection integral may change by multiples of $2\pi$ upon

singular gauge transformations, while the Berry flux is fully gauge-invariant, Eq. (2) only holds modulo $2\pi$. We rewrite the Stokes' theorem in terms of the Wilson line

$$\mathcal{W}_{\mathcal{C}} = \exp\left(i \int_{\mathcal{C}} \mathcal{A}\, d\mathbf{k}\right), \tag{3}$$

where the exponent is path-ordered along $\mathcal{C}$. Under a gauge transformation of the occupied wavefunctions $|n\mathbf{k}\rangle \rightarrow \sum_m |m\mathbf{k}\rangle U_{mn}(\mathbf{k})$, the Wilson line transforms according to $\mathcal{W}_{\mathcal{C}} \rightarrow U^{\dagger}(\mathbf{k}_f)\mathcal{W}_{\mathcal{C}} U(\mathbf{k}_i)$, where $\mathbf{k}_i$ and $\mathbf{k}_f$ are the initial and final points of $\mathcal{C}$. If the path $\mathcal{C}$ is closed, Eq. (3) defines a Wilson loop, whose eigenvalues are gauge invariant, while the Wilson line spectrum is gauge dependent [15]. Substituting Eq. (3) into Eq. (2) yields

$$\int_{\text{IBZ}} \text{tr}\,\mathcal{F}\, d\mathbf{k}^2 + i \log \det \mathcal{W}_{\partial\text{IBZ}} = 0 \quad \text{mod } 2\pi. \tag{4}$$

This identity defines the discrete quantity that we use for the topological invariant. However, without applying symmetry constraints, Eq. (4) carries no information due to the gauge ambiguity of $2\pi$.

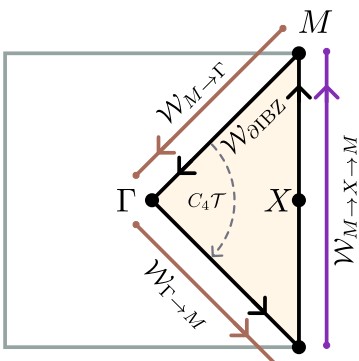

Figure 2: The irreducible Brillouin zone (yellow) spans the Brillouin zone together with its $C_4\mathcal{T}$-images. Its boundary is constrained by $C_4\mathcal{T}$-symmetry (grey dashed arrow), so we split the Wilson loop $\mathcal{W}_{\partial\text{IBZ}}$ (black arrows) at the high symmetry momenta $\Gamma$ and $M$. Two $C_4\mathcal{T}$-equivalent Wilson lines, $\mathcal{W}_{\Gamma \rightarrow M}$ and $\mathcal{W}_{M \rightarrow \Gamma}$, (brown arrows), and a Wilson loop, $\mathcal{W}_{M \rightarrow X \rightarrow M}$, (purple arrow), compose the resulting path.

To resolve the gauge ambiguity, we split the Wilson loop $\mathcal{W}_{\partial\text{IBZ}}$ into symmetry-constrained parts. Specifically, we consider the Wilson lines from $\Gamma \rightarrow M$, $M \rightarrow X \rightarrow M$, and $M \rightarrow \Gamma$. Because all of these Wilson lines start and end at $C_4\mathcal{T}$-invariant momenta, we constrain their spectrum using the $\Theta$ operator. We define the dressed Wilson line determinant as

$$\widetilde{\det}\mathcal{W}_{\mathcal{C}} = \text{pf}^{-1} w(\mathbf{k}_f) \det \mathcal{W}_{\mathcal{C}}\, \text{pf}\, w(\mathbf{k}_i). \tag{5}$$

Here $\mathbf{k}_i$ and $\mathbf{k}_f$ must be either $\Gamma$ or $M$, the start and end points of the path $\mathcal{C}$, respectively. The antisymmetric overlap matrix $w(\mathbf{k})$ is the projection of the $\Theta$ operator on the occupied states $w_{mn}(\mathbf{k}) = \langle n\mathbf{k}|\Theta|m\mathbf{k}\rangle$, and pf is the Pfaffian. An alternative approach to define the dressed Wilson line is to use the generalized Pfaffian [16] of the $C_4\mathcal{T}$ overlap matrix $w'_{mn}(\mathbf{k}) = \langle n\mathbf{k}|C_4\mathcal{T}|m\mathbf{k}\rangle$, or the Pfaffian of the antisymmetrized overlap matrix [17], $(w' - w'^T)/2$. Due to the gauge transformation property $w(\mathbf{k}) \rightarrow U^{\dagger}(\mathbf{k})w(\mathbf{k})U^*(\mathbf{k})$, and the identity $\text{pf}(CAC^T) = \det(C)\,\text{pf}(A)$, the dressed

Wilson line determinant is gauge invariant. Furthermore, because the three paths combine into the contour of the IBZ,

$$\det \mathcal{W}_{\partial \text{IBZ}} = \widetilde{\det} \mathcal{W}_{\Gamma \to M} \, \widetilde{\det} \mathcal{W}_{M \to X \to M} \, \widetilde{\det} \mathcal{W}_{M \to \Gamma} = \widetilde{\det}^2 \mathcal{W}_{\Gamma \to M} \, \widetilde{\det} \mathcal{W}_{M \to X \to M}. \tag{6}$$

Here we used that $M \to \Gamma$ is the $C_4 \mathcal{T}$-image of $\Gamma \to M$, and therefore $\widetilde{\det} \mathcal{W}_{\Gamma \to M} = \widetilde{\det} \mathcal{W}_{M \to \Gamma}$. Finally, we recognize that the initial and final momenta of $M \to X \to M$ are the same, so that $\widetilde{\det} \mathcal{W}_{M \to X \to M} = \det \mathcal{W}_{M \to X \to M}$ is the Wilson loop determinant. The $C_2$-invariance of this path further constrains the Wilson loop determinant

$$\det \mathcal{W}_{M \to X \to M} = \prod_{n \in \text{occ}} \frac{\zeta_n(X)}{\zeta_n(M)} = \prod_{n \in \text{occ}} \zeta_n(X) \equiv \delta, \tag{7}$$

where $\zeta_n(\mathbf{k}) = \pm i$ is the eigenvalue of the operator $C_2$ of the $n$th occupied band at $C_2$-invariant momenta [18]. For the second equality, we observe that the product of $C_2$ eigenvalues at the $M$ point is always trivial due to the Kramers-like degeneracy. As a consequence, the Wilson loop determinant is equal to the $C_2$ symmetry indicator $\delta = \pm 1$.

To construct the invariant we substitute Eq. (6) into Eq. (4), subtract the logarithm of Eq.(7), and obtain

$$\nu = \frac{1}{\pi} \left[ \int_{\text{IBZ}} \text{tr} \, \mathcal{F} \, d\mathbf{k}^2 + 2i \log \widetilde{\det} \mathcal{W}_{\Gamma \to M} \right] \quad \text{mod } 4. \tag{8}$$

This is our main result. The invariant is defined modulo 4 because each dressed Wilson line determinant is well-defined modulo $2\pi$. The invariant is also quantized to integer values and it stays constant as long as the spectrum is gapped. However, at this point, the relation between the invariant and the different phases is not yet established.

To show that $\nu$ distinguishes the four atomic insulators shown in Fig. 1, we test it by applying it to the corresponding phases. We construct the Hamiltonians of each atomic limit from coupled spinful $p$-type orbitals. The orbitals are located at the center of the unit cell and transform into each other under $C_4$ rotations. Using the standard representation of spin 1/2, yields

$$C_4 \mathcal{T} = \tau_y e^{-i\sigma_z \pi/4} \sigma_y \mathcal{K}, \tag{9}$$

where $\tau_i$ are the Pauli matrices in orbital space in the basis $p_x$, $p_y$, $\sigma_i$ are the Pauli matrices in spin space, and $\mathcal{K}$ is complex conjugation. In the trivial limit, we couple opposite spins within each orbital in the unit cell to obtain a spin singlet located on each atom, as shown in Fig. 3(a). In the obstructed atomic limits the spins are localized in between unit cells, hence we couple opposite spins from the same orbital type that belong to neighboring unit cells, as shown in Fig. 3(b-d). Specifically, to couple opposite spin-polarized states in all the atomic limits, we use the operator

$$S(\mathbf{r}, \delta \mathbf{r}, \mathbf{\Omega}) = |\mathbf{r} + \delta \mathbf{r}, \mathbf{\Omega}\rangle \langle \mathbf{r}, -\mathbf{\Omega}| + |\mathbf{r}, -\mathbf{\Omega}\rangle \langle \mathbf{r} + \delta \mathbf{r}, \mathbf{\Omega}|, \tag{10}$$

where $|\mathbf{r}\rangle$ is the state localized at a unit cell with coordinates $\mathbf{r}$, $\delta \mathbf{r}$ is the displacement between the coupled unit cells, and $|\mathbf{\Omega}\rangle$ is a spin oriented in the $xy$-plane along the direction $\mathbf{\Omega}$. We require $|\mathbf{\Omega}\rangle = -iC_2 |-\mathbf{\Omega}\rangle$. This guarantees that the occupied eigenstate of $S$ is a $+i$ eigenstate of a $C_2$ rotation around $\mathbf{r} + \delta \mathbf{r}/2$ or, in other words, it is a $+z$-spin located at $\mathbf{r} + \delta \mathbf{r}/2$. Without loss of generality, we choose $\mathbf{\Omega} = \hat{y}$. Since the occupied states in the atomic limits are $z$-aligned spins of different orbital characters, as shown in Fig. 3, we use the projector on the two orthogonal $p$-orbitals

$$T_{\pm} = \frac{1}{2} \left[ 1 \pm (\tau_z \cos \phi + \tau_x \sin \phi) \right], \tag{11}$$

to couple spins within orthogonal orbitals. Here $\phi$ is an arbitrary orbital polarization in $xy$-plane that we choose as $\phi = 0$. The atomic limits shown in Fig. 3 are then given by the Hamiltonians

$$H^{(\nu)} = \sum_{r} \left[ T_+ S(\boldsymbol{r}, \delta\boldsymbol{r}_\nu, \boldsymbol{\Omega}) - T_- S(\boldsymbol{r}, \hat{\boldsymbol{z}} \times \delta\boldsymbol{r}_\nu, \hat{\boldsymbol{z}} \times \boldsymbol{\Omega}) \right] \tag{12}$$

for $\nu = 0, 1, 2, 3$, where $\delta\boldsymbol{r}_0 = 0$, $\delta\boldsymbol{r}_1 = \hat{x}$, $\delta\boldsymbol{r}_2 = \hat{x} + \hat{y}$, and $\delta\boldsymbol{r}_3 = \hat{y}$. We recognize that $H^{(3)} = -H^{(1)}$, in agreement with the atomic limits in Fig. 1(c-d) being the time-reversed images of each other.

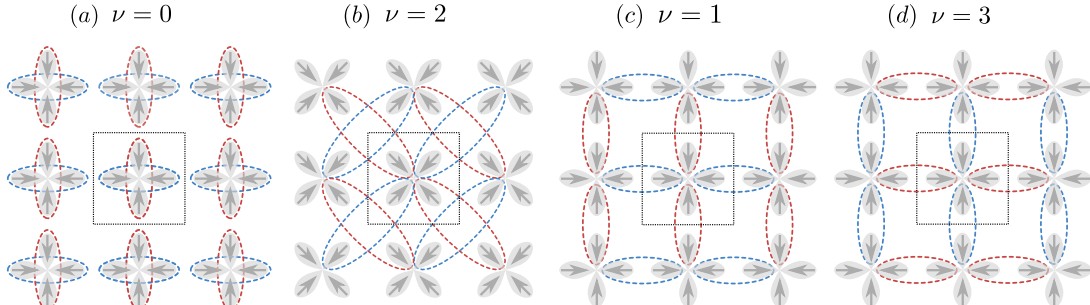

(a) $\nu = 0$       (b) $\nu = 2$       (c) $\nu = 1$       (d) $\nu = 3$

Figure 3: Construction of the atomic limits from coupling opposite spins within $p_x$ and $p_y$ orbitals (orthogonal pairs of arrows) located at the center of the unit cell (square). Positive couplings (red ellipses) result in $+z$-oriented spins, while negative ones (blue ellipses) result in $-z$-oriented spins.

To confirm that the invariant is quantized and that it only changes under gap closing transitions, we construct a model of a $C_4\mathcal{T}$-invariant planar magnetic insulator with no other symmetries. Its Hamiltonian interpolates between the four atomic limits and contains additional onsite terms breaking extra symmetries

$$H = \alpha H^{(0)} + \beta H^{(1)} + \gamma H^{(2)} + \sum_{i=4}^{N=23} \lambda_i H^{(i)}. \tag{13}$$

Here $\alpha$, $\beta$, and $\gamma$ are the weights of the atomic limit Hamiltonians, and $\lambda_i$ are the amplitudes of the 19 other $C_4\mathcal{T}$-invariant onsite terms $H^{(i)}$, which we generate using Qsymm [19]. We choose $\alpha \in [0, 1]$, $\beta \in [-1, 1]$, and $\gamma = 1 - \alpha - |\beta| \in [0, 1]$, such that the $\nu = 3$ atomic limit is included in the negative range of $\beta$. We use Adaptive [20] to sample Hamiltonians whose energy gaps we find via numerical minimization. We numerically compute the invariant using occupied band projectors to discretize the Wilson lines [15] over a Brillouin zone grid of $20 \times 20$ momenta, where we choose the upper right quadrant of the Brillouin zone as the IBZ for simplicity. Choosing a different IBZ does not change the invariant because all the possible IBZ are smoothly connected to one another, but Eq. (8) only takes integer values. The phase diagram in Fig. 4(a) shows the interpolation between the atomic limits without additional $C_4\mathcal{T}$-invariant terms ($\lambda_i = 0$), where the conservation of the orbital polarization protects the gapless region. To break additional symmetries we set $\lambda_i = 0.08$ and obtain the phase diagram in Fig. 4(b). Our results confirm that $\nu$ is quantized and it labels the four phases, each adiabatically connected to an atomic limit. The phase diagrams show transitions with $\nu$ changing by $\pm 1$ or by 2, where the former are accompanied by a gap closing at the $X$ point that changes the $C_2$ indicator of Eq. (7). If the gap closes at a different momentum, the transition is overlooked by the $C_2$ indicator, but not by the invariant $\nu$, which changes by 2.

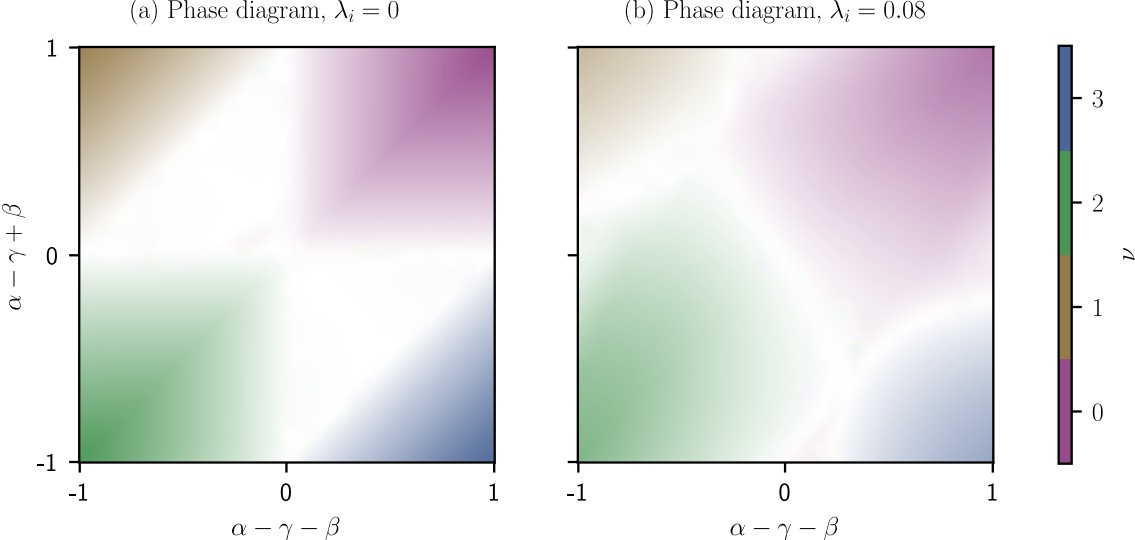

Figure 4: Phase diagram of the $C_4\mathcal{T}$ model without (a) and with (b) additional symmetry breaking terms. The invariant is quantized and it distinguishes the four atomic limits at the corners of panel (a). Away from the atomic limits, the invariant stays quantized and the energy gap becomes smaller (increasing transparency). Across phase transitions the invariant changes by $\pm 1$ and 2 via energy gap closings (white) at $X$ or a different momentum, respectively.

The construction of the atomic limits shown in Fig. 3, suggests that the boundary charge of each phase depends on the lattice termination. To determine the bulk-boundary correspondence of the phases, we compute the charge density in square and rhombus-shaped lattices, such that the termination cuts a different number of bonds along the boundary. We place the Fermi level inside the gap, $E_F = 0.15$, and compute the deviation of charge per unit cell from $2e$—the charge density at half-filling. We use Kwant [21] to construct square and rhombus geometries of $L^2 = 7 \times 7$ and $L^2 = 9 \times 9$ unit cells, respectively, for each phase. We choose the amplitudes $\alpha$, $\beta$, and $\gamma$ as 0.6, 0.2, 0.2, where the biggest amplitude determines the phase, and set the symmetry breaking terms $\lambda_i = 0.01$. While the trivial phase lacks boundary modes in either geometry, the obstructed phases localize $1/2e$ per bond cut by the boundary, as shown in Fig. 5.

In summary, we derived an invariant that distinguishes the inequivalent atomic insulating phases of the $p4'$ planar magnetic group. We applied Stokes' theorem to the Berry connection over the irreducible Brillouin zone and exploited the $C_4\mathcal{T}$ symmetry to constrain the phase contributed by the open Wilson lines and a Wilson loop. While the Wilson loop contribution is equal to that of the eigenvalues of $C_2$ at the $X$ point, and is therefore insufficient to distinguish all phases, the Berry flux and the Wilson lines complete the $\mathbb{Z}_4$ invariant. Alternatively, our invariant is equivalent to the vorticity of the Pfaffian of the overlap matrix $w$ over half a Brillouin zone modulo 4, similar to the $\mathbb{Z}_2$ invariant Refs. [6,11–14], although this formulation has the disadvantage of requiring a smooth gauge. We constructed models away from the atomic limits and found that the obstructed

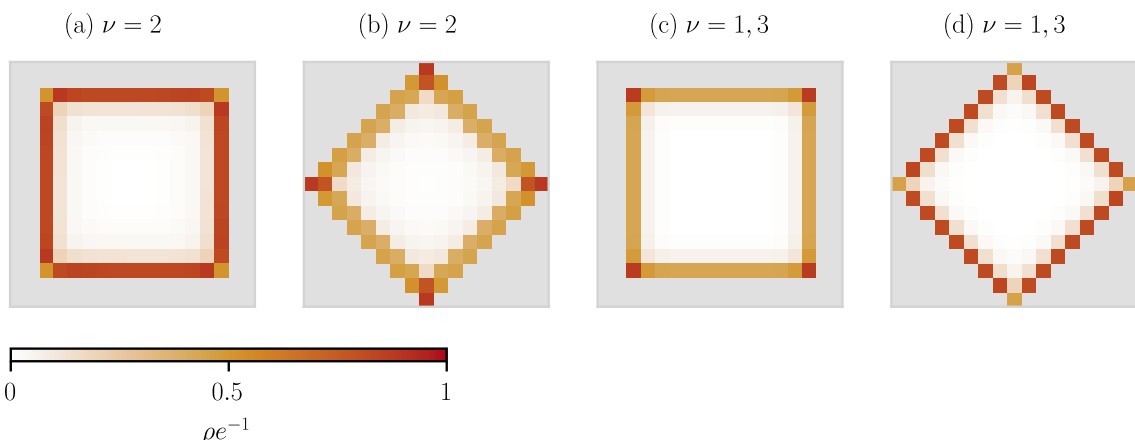

Figure 5: The local charge density of the four phases depends on the termination of the lattice, in agreement with the construction procedure. In the obstructed phase $\nu = 2$, $1e$ per unit cell localizes at the edge, and $1/2e$ at the corner in a square lattice (a), and vice versa in a rhombus geometry (b). In the obstructed phases $\nu = 1$ and $\nu = 3$, $1/2e$ per unit cell localizes at the edge, and $1e$ at the corner in a square lattice (c), and vice versa in a rhombus geometry (d).

phases may undergo both transitions changing the invariant by $\pm 1$ or 2, depending on whether the energy gap closes at $X$ or a different momentum.

Our work confirms that the Berry phase alone is insufficient to classify obstructed atomic insulators protected by magnetic space groups. Applying the approach presented here to other magnetic groups will allow to complete the construction of topological invariants that distinguish all obstructed atomic insulating phases. Different values of the topological invariant may exist in neighboring domains of altermagnets [22], where the $\mathcal{T}$ symmetry is spontaneously broken. The bulk-boundary correspondence will then govern the spin and charge properties of the domain walls, and therefore influence their energetic stability.

## Acknowledgements

We are grateful to I. C. Fulga, K. K. Pöyhönen, A. Lau, H. Spring, A. L. R. Manesco, and F. Schindler for fruitful discussions.

## Data availability

The code used to produce the reported results is available on Zenodo [23].

**Author contributions**    D. V. initiated the project and identified the initial formulation of the invariant. A. V. implemented the computation of the invariant with input from D. V., validated it, studied the bulk boundary correspondence, and wrote the initial summary with D. V. D. V. and A. R. A. constructed the atomic limits model. I. A. D. identified the relevant literature, and devel-

oped the idea to its final form with D. V. and A. R. A. I. A. D. wrote the final implementation of the code with input from A. R. A., and wrote the manuscript with input from D. V. and A. R. A. The project was managed by A. R. A. with contributions from D. V.

**Funding information** This work was supported by the Netherlands Organization for Scientific Research (NWO/OCW) as part of the Frontiers of Nanoscience program, and an NWO VIDI grant 016.Vidi.189.180. D. V. was supported by NWO VIDI Grants 680-47-537, the Swedish Research Council (VR), and the Knut and Alice Wallenberg Foundation.

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
