# Peer review of "Pfaffian invariant identifies magnetic obstructed atomic insulators"

_SciPost Physics_

## Round 2 · Referee Report · Frank Schindler (Referee 1) · 2022-12-29

Strengths

1- solves an open problem 2- has potential for follow-up work 3- is clearly written

Weaknesses

1- only treats a single symmetry class

Report

This paper derives a topological invariant for two-dimensional obstructed atomic insulators with C4T symmetry. Symmetry indicator invariants only give a partial classification of insulators in this symmetry class. The authors solve this problem by formulating an invariant in terms of a Berry curvature integral and a Wilson line dressed with Pfaffian terms. The resulting invariant uniquely distinguishes all four inequivalent atomic insulators with C4T symmetry. Moreover, unlike other Pfaffian-based invariants, computing the invariant does not require a smooth gauge of Bloch states.

Since the paper solves an open problem it is well suited for publication in SciPost Physics. Moreover, the general idea seems easily extendable to other symmetry classes. Here my only complaint is that the authors have not attempted such a generalization, and I would ask them to at least comment on it. As the paper is otherwise well written, I am happy to recommend publication once the minor concerns listed below are addressed.

Requested changes

  • The term $\text{mod } 4$ in Eq. (8) needs additional explanation. The statement in the text "The invariant [...] is well-defined modulo 4 due to the gauge-fixing procedure" is somewhat cryptic.

  • It would be helpful to explain the origin of the term $\text{mod } 2\pi$ in Eq. (2) as this term is normally absent in Stokes' theorem.

  • Below Eq. (13) it is stated "we choose the upper right quadrant of the Brillouin zone as the IBZ for simplicity". Is this really allowed when evaluating Eq. (8)? The lower half of the IBZ highlighted in Fig. 2 is rotated into the upper right quadrant by C4T. Since C4T involves time-reversal, it flips the sign of Berry curvature. Hence the Berry curvature integral over the upper right quadrant will in general be different from the integral over the IBZ of Fig. 2.

  • Could there be a sign error in the second term of Eq. (4)? Using $\log \det = \mathrm{tr} \log$ results in $+ \int \mathrm{tr} A$, not $- \int \mathrm{tr} A$ as would be required from Eq. (2).

  • The invariants $\delta$ and $\nu$ appear in Fig. 1 without explanation, and much earlier than their definition in the text.

  • I do not agree with the statement "...because the phases within each pair are equivalent up to a fractional lattice vector translation, the symmetry indicators only provide an incomplete topological classification". For instance, take inversion symmetry in 1D. The two atomic insulator phases (with Wannier centers at the 1a or 1b Wyckoff positions) are equivalent up to a half lattice vector translation, yet they are fully resolved by symmetry indicators.

---

## Round 2 · Referee Report · Kai Sun (Referee 2) · 2023-1-18

Strengths

  1. Thorough analysis of an very interesting problem
  2. Very clear presentation

Weaknesses

  1. It would help further strengthen the manuscript, if more discussion about potential experimental impacts can be added

Report

This work studies topological classification of two-dimensional insulators with C4T symmetry. Utilizing the pfaffian defined through an anti-unitary operator, the authors identified a Z4 topological index, which perfect match with 4 different types of obstructed atomic insulators. The manuscript further verified this Z4 classification by examining the edge/corner charge, gap closing via adiabatic deformation, and the construction of the atomic limits. This study provide a new and efficient way to analysis and classify topological states for this symmetry family, and the results of the study are intriguing and timely. The mathematical proof and technique are clearly presented. I believe this work is suitable for publication in SciPost.

Below are some minor thoughts for the authors to consider, mainly about the background and potential impacts/implications of this work:
(1) I believe that the C4T symmetry is compatible with altermagnetism (also known as nematic-spin-nematic). So one potential impact of this study is to provide a topological classification for altermagnetic materials, which could be a fun topic, considering the recent interests in these systems.

(2) “On the other hand, because the phases within each pair are equivalent ...” As pointed out in another review report, this sentence needs some further clarification. Fractional lattice vector translation itself doesn’t seem to provide a sufficient condition for the symmetry indicators to be identical. I believe that it would need to involve more details about the space group symmetry, e.g., location of the rotation center, to fully clarify this statement here.

(3) Related with altermagnetism, because the C4 and T symmetry is expected to be spontaneously broken in these systems, an altermagnetic system should have two types of domains (related to each other by a C4 or T transformation). Along this line of thinking, the result reported in this study seems to indicate that there are 3 different types of altermagnetic insulators: (a) both domains have nu=0, (b) both domains have nu=2, (c) one domain has nu=1 and the anti domain has nu=3. If this is the case, Fig 5 provide a potential way to distinguish these different families of altermagnetism (via edge/corner states). In addition, it would probably be interesting to look at the domain walls for family (c), between nu=1 and nu=3, and see if the domain support localized spin/charge excitations. For altermagnetic materials in this family (c), such domain walls should automatically arise as the temperature is cooled down below the transition temperature. Thus, such domain wall states could potentially be interesting experimental signature for this topological family.

---

## Round 3 · Referee Report · Frank Schindler · 2023-6-5

Report
The authors have addressed my previous concerns in a satisfactory manner and I am happy to recommend publication.

---

## Round 3 · Referee Report · Kai Sun · 2023-6-14

Report
All concerns raised previously have been fully addressed by the authors, and I am delighted to recommend the publication of this manuscript.

---

## Round 3 · Author Response

Resubmission letter
We thank the referees for their feedback and for their overall positive evaluation. Below we list the detailed response to the referee inquiries, followed by the list of changes in the manuscript.
Response to referee 1
-
The term mod 4 in Eq. (8) needs additional explanation. The statement in the text "The invariant [...] is well-defined modulo $4$ due to the gauge-fixing procedure" is somewhat cryptic.
We have clarified the sentence by changing it to "The invariant is defined modulo $4$ because each dressed Wilson line determinant is well-defined modulo $2 \pi$."
-
It would be helpful to explain the origin of the term $mod \; 2 \pi$ in Eq. (2) as this term is normally absent in Stokes' theorem.
We have added the following statement "Since the Berry connection integral may change by multiples of $2\pi$ upon singular gauge transformations, while the Berry flux is fully gauge-invariant, Eq.(2) only holds modulo $2\pi$."
-
Below Eq. (13) it is stated "we choose the upper right quadrant of the Brillouin zone as the IBZ for simplicity". Is this really allowed when evaluating Eq. (8)? The lower half of the IBZ highlighted in Fig. 2 is rotated into the upper right quadrant by $C_4 \mathcal{T}$. Since $C_4 \mathcal{T}$ involves time-reversal, it flips the sign of Berry curvature. Hence the Berry curvature integral over the upper right quadrant will in general be different from the integral over the IBZ of Fig. 2.
The sign of the Berry curvature indeed flips upon the action of $C_4 \mathcal{T}$, however Equation (8) does not depend on the choice of the IBZ. Specifically, the IBZ in Fig. 2 deforms into the upper right quadrant of the BZ by a smooth deformation that changes both the Berry flux and the Wilson line continuously. This means that in the numerical implementation we consider the Wilson loop $M \to X \to M$ and twice the Wilson line $M \to Y \to \Gamma$. Since Eq. (8) only takes integer values, its value cannot change upon smooth deformations of the IBZ. Therefore, using either IBZ gives the same invariant, which we have confirmed numerically. We only use the right upper quadrant to simplify the integration over a square grid in the Brillouin zone. To clarify this we have added a sentence "Choosing a different IBZ does not change the invariant because all the possible IBZ are smoothly connected to one another, but Eq.(8) only takes integer values."
-
Could there be a sign error in the second term of Eq. (4)? Using $\text{log} \; \text{det} = \text{tr} \; \text{log}$ results in $\int \text{tr} \mathcal{A}$ , not $-\int \text{tr} \mathcal{A}$ as would be required from Eq. (2).
We thank the referee for finding a typo in our manuscript. In the previous version, Eq. (4) was consistent with Figure 2, but the Wilson loop's path went against the convention of using counter-clockwise paths in Stokes' theorem. In the current version, we have reversed the path of Figure 2, and changed the sign in Eq. (4), so these agree with Eq.(2-3). Since the inconsistency was only present in the text and not in the code, our results remain the same.
-
The invariants $\delta$ and $\nu$ appear in Fig. 1 without explanation, and much earlier than their definition in the text.
We have modified paragraphs 2 and 3 to introduce $\delta$ and $\nu$ before Figure 1.
-
I do not agree with the statement "...because the phases within each pair are equivalent up to a fractional lattice vector translation, the symmetry indicators only provide an incomplete topological classification". For instance, take inversion symmetry in 1D. The two atomic insulator phases (with Wannier centers at the 1a or 1b Wyckoff positions) are equivalent up to a half lattice vector translation, yet they are fully resolved by symmetry indicators.
We have modified the sentence to "However, even the full set of symmetry indicators only provides an incomplete topological classification: phases $\nu=0$ and $\nu=2$ have identical representation content at every high-symmetry momentum."
Response to referee 2
We thank the referee for the interesting suggestions about altermagnetism and the experimental relevance of our work. We believe that addressing these questions would require a separate investigation, and therefore we leave them to future work. We have addressed the referee's suggestion in the concluding remarks of the new version.
-
“On the other hand, because the phases within each pair are equivalent ...” As pointed out in another review report, this sentence needs some further clarification. Fractional lattice vector translation itself doesn’t seem to provide a sufficient condition for the symmetry indicators to be identical. I believe that it would need to involve more details about the space group symmetry, e.g., location of the rotation center, to fully clarify this statement here''
We have modified the sentence to "However, even the full set of symmetry indicators only provides an incomplete topological classification: phases $\nu=0$ and $\nu=2$ have identical representation content at every high-symmetry momentum."

---

## Round 3 · List of Changes

- Fixed a minus sign typo in Eqs. 4 and 8
- Fixed the orientation of the IBZ in Fig. 2
- Introduced $\nu$ and $\delta$ before Fig. 1
- Clarified the origin of mod 4 in Eq. 8
- Clarified the IBZ used for numerical integration
- Complemented the conclusion with altermagnets
We also attach a PDF file with the changes highlighted:
https://surfdrive.surf.nl/files/index.php/s/rrLFIVVZr2k0dTl

You are currently on this page

---

## Editorial Decision

editorial_decision: